# OpenReview forum: "When AI Agents Collude Online: Financial Fraud Risks by Collaborative LLM Agents on Social Platforms"
_ICLR.cc/2026/Conference — ICLR 2026 Poster_

### Official Review · Reviewer_U9v1 · 2025-10-27

**Soundness:** 3
**Presentation:** 3
**Contribution:** 3
**Rating:** 6
**Confidence:** 4

**Summary:**

This paper studies the risks of **financial fraud collusion** in multi-agent systems (MAS) driven by LLMs. The authors propose *MultiAgentFraudBench*, a benchmark covering the **entire fraud lifecycle** (public lure, private trust-building, final transfer), and evaluate 16 mainstream LLMs under 21 scenarios. Results show that stronger models have higher fraud success rates, and that **collusion channels significantly amplify harm**. The paper also proposes preliminary mitigation strategies, including *monitor agents* that detect and block malicious agents, and *group resilience* mechanisms encouraging benign agents to share warnings.

**Strengths:**

* **Originality**: Introduces one of the first systematic benchmarks to study financial fraud collusion in MAS.
* **Quality**: Experiments are comprehensive, covering 16 LLMs and multiple fraud scenarios. Ablation studies (scale, ratios, collusion vs. non-collusion) add credibility.
* **Clarity**: Writing is mostly clear, with strong use of figures/tables to illustrate fraud dynamics.
* **Importance**: Highlights a timely and socially important problem—the potential misuse of MAS for coordinated fraud—and provides a foundation for further work in mitigation.

**Weaknesses:**

* Scenarios are fully synthetic (LLM-generated posts), which weakens external validity. More discussion or comparison with real-world fraud cases is needed.
* The alignment failure is tested by forcing malicious system prompts, which may exaggerate risks compared to real-world deployment.
* Proposed defenses (debunking, banning, resilience) are interesting but technically shallow. Stronger contributions could involve mechanism design at the agent or system level.
* I recommand you change your bench's name, it's too big as you mainly discuss financial fraud.

**Questions:**

* Can you clarify how realistic they expect *MultiAgentFraudBench* to be when extrapolated to real-world fraud ecosystems?
* Could benign agents develop **emergent self-defense behaviors** without being explicitly prompted, instead of relying only on prescribed resilience mechanisms?
* How would fraud success interact with recommender system biases in real platforms? Could this be modeled more systematically?

---

> ### Author Response · Authors · 2025-11-23
> **Response to Reviewer U9v1 (Part 1/3)**
>
> Thank you for your great efforts in reviewing this paper and for recognizing our work as a "timely and socially important" problem, identifying it as "one of the first systematic benchmarks" in this field, and appreciating the "comprehensive" nature of our experiments. We will try our best to answer all your questions. Please let us know if you still have further concerns, or if you are not satisfied with the current responses, so that we can further update the response ASAP.
>
> ---
>
> **Q1**: Scenarios are fully synthetic (LLM-generated posts), which weakens external validity. More discussion or comparison with real-world fraud cases is needed.
>
> **A1**: Thank you for this helpful suggestion! We followed your recommendation and compared the fraud trajectories of our agents with real-world fraud cases [1,2,3]. As shown in Appendix G.2, we found that the trajectory of a successful scam conducted by our malicious agents is highly consistent with real-world fraud patterns. These include posting scam messages to attract victims, building trust through private chats, and finally stealing money through transfers or malicious links.
>
> For example, in the case of an NFT Whitelist and Gas Fee Scam, our simulated User 108 impersonates an NFT platform staff member and tells User 41 that they have been selected for a “VIP whitelist’’ in a charity-themed NFT auction. User 41 is asked to pay a refundable deposit and several small payments for account verification and whitelist maintenance. Before the NFT is released, User 108 requests a final Ethereum gas fee for minting. After receiving the cryptocurrency transfer, User 108 stops all communication. This fraud trajectory closely mirrors real-world NFT scams. Appendix G.3 includes more case comparisons. These analyses strengthen the realism of our benchmark. Thank you again for the suggestion. We have updated these results in the revised version.
>
> We would also like to clarify our motivation for using synthetic posts. First, our posts were generated based on 21 real fraud scenarios, and we provided random user profiles during generation to make the content more targeted and realistic. Second, **this approach allows us to align with current and foreseeable attack patterns.** Several studies have shown that real-world criminals are already outsourcing content creation to LLMs, including fake investment advice and phishing messages [4,5]. We will add this discussion to the camera-ready paper.
>
> References:
>
> 1. Africa Check. Analysis: About the Money? How to Spot a Scam Conference.
> https://africacheck.org/fact-checks/blog/analysis-about-money-how-spot-scam-conference
>
> 2. Heart & Soul Whisperer. Case Studies of Art Scams.
> https://www.heartandsoulwhisperer.com.au/case-studies-of-art-scams
>
> 3. Elliptic. NFTs and Financial Crime Report. 2022.
> https://www.elliptic.co/hubfs/NFT%20Report%202022%20-full.pdf
>
> 4. Fingerprint. Large Language Models (LLMs) in Fraud and Malware: A Practical Guide.
> https://fingerprint.com/blog/large-language-models-llm-fraud-malware-guide/?utm_source=chatgpt.com
>
> 5. Microsoft Security. AI vs. AI: Detecting an AI-Obfuscated Phishing Campaign. 2025.
> https://www.microsoft.com/en-us/security/blog/2025/09/24/ai-vs-ai-detecting-an-ai-obfuscated-phishing-campaign/
>
> ---
>
> **Q2**: The alignment failure is tested by forcing malicious system prompts, which may exaggerate risks compared to real-world deployment.
>
> **A2**: Thank you! We understand your concern. However, our setup is actually based on how malicious actors would deploy harmful agents in real-world conditions. As shown in Figure 2, we only use a natural-language system prompt to describe the fraud task: “You are a member of a malicious group on Twitter, where both factual and deceptive content circulate. Your group’s goal is to maximize the effectiveness of fraudulent schemes...” We do NOT apply any jailbreak strategies to force the agents into risky behavior, and we do not retry when an agent refuses to follow the malicious instruction.
>
> In our experiments, we found that except for Llama-3.1-405B, which often refused by selecting “do nothing,” all other models rarely refused (line 262). This indicates that **current alignment methods focus on isolated Q&A settings and fail to generalize to our interactive, agent-based environment.** It also suggests that existing open- and closed-source agents can be easily exploited by real-world malicious users, even if those users do not know advanced jailbreak strategies. This creates a realistic risk of large-scale amplification.
>
> We hope that our benchmark can motivate the community to study safety alignment for agents operating in real-world social environments.

---

> ### Author Response · Authors · 2025-11-23
> **Response to Reviewer U9v1 (Part 2/3)**
>
> ---
>
> **Q3**: Proposed defenses (debunking, banning, resilience) are interesting but technically shallow. Stronger contributions could involve mechanism design at the agent or system level.
>
> **A3**: Thank you for your question! We acknowledge that our two prompt-based defense strategies are preliminary attempts. However, we would like to highlight our contributions in mitigation:
>
> 1. Introducing a new idea for defense is our contribution. We are the first to introduce the concept of group resilience to mitigate social risks caused by AI agents. Our experiments show that spontaneous information spreading and resistance within a group can improve the robustness of the entire group against attacks. This approach does not require extra models or guardrails, so the community can continue to explore this promising direction.
>
> 2. Providing a multi-angle and systematic evaluation of defenses is also our contribution. Our paper mainly builds a benchmark to reveal fraud risks caused by collaborative agents. To complete the loop, we implement classical debunking and also propose two of our own methods. These methods cover the post level, the individual level, and the group level. We hope this systematic evaluation can guide future follow-up work in the community.
>
> ---
>
> **Q4**: I recommend you change your bench's name, it's too big as you mainly discuss financial fraud.
>
> **A4**: Thank you! This is a very helpful suggestion. We have already changed our name from *MultiAgentFraudBench* to *MultiAgentFinancialFraudBench* in our revised paper.
>
> ---
>
> **Q5**: Can you clarify how realistic they expect MultiAgentFraudBench to be when extrapolated to real-world fraud ecosystems?
>
> **A5**: Sure! MultiAgentFraudBench aims to be realistic when extrapolated to real-world fraud ecosystems. Our benchmark includes three layers of design to ensure authenticity:
>
> 1. **End-to-end realistic fraud pipeline**: our benchmark simulates fraud trajectories that closely match real-world cases. The process includes spreading fraudulent posts to attract victims, building trust through private messages, and finally deceiving them through money transfers or malicious links. After being scammed, “good” users may also warn the community, forming a dynamic and realistic fraud simulation pipeline.
>
> 2. **Diverse and realistic fraud content and scenarios**: We generate posts based on 21 real fraud scenarios. During generation, we also provide (1) a random user profile, (2) real fraud post examples, and (3) scenario-specific fraud strategies. These elements make the generated content more targeted and more realistic.
>
> 3. **Realistic threat model and large-scale simulation capability**: First, to make our simulation faithfully mirror real-world conditions, we construct a threat model to define the simulation boundary. This model includes realistic ratios of malicious to benign agents, comparable knowledge and activity levels, and freedom to interact through standard social media actions. Moreover, our simulation environment integrates a recommendation system and supports large-scale simulations with thousands or even millions of agents. In contrast, prior work on multi-agent collusion has mainly used small-scale or simplified environments [1,2].
>
> References:
>
> 1.  Mathew et al. Hidden in Plain Text: Emergence & Mitigation of Steganographic Collusion in LLMs. arXiv:2410.03768, 2024.
>
> 2.  Motwani et al. Secret Collusion Among AI Agents: Multi-agent Deception via Steganography. NeurIPS 37, 2024.
>
> ---

---

> ### Author Response · Authors · 2025-11-23
> **Response to Reviewer U9v1 (Part 3/3)**
>
> ---
>
> **Q6**: Could benign agents develop emergent self-defense behaviors without being explicitly prompted, instead of relying only on prescribed resilience mechanisms?
>
> **A6**: Yes, absolutely. This is a very insightful point. We did observe that benign agents can develop emergent self-defense behaviors when they realize, in their chain-of-thought, that the current context contains fraud-related information.
>
> (1) As shown in Appendix F.2, agents sometimes take self-defense actions such as “block_user” and “report_user”, even though these actions are not explicitly coded in the action space.
>
> (2) They also post security alerts. For example, a benign agent may write: “Important update: Received an official security alert regarding a scam attempt by User 103. Please stay vigilant and report any suspicious activities. #StaySafeOnline.”
>
> We believe that strengthening agents’ self-awareness when they face sensitive or malicious content—rather than relying only on predefined resilience mechanisms—is a promising direction for future work.
>
> ---
>
> **Q7**: How would fraud success interact with recommender system biases in real platforms? Could this be modeled more systematically?
>
> **A7**: This is a very important question. We agree that the recommender system plays a key role in our simulation. We follow the technical report of X to build our recommender system [1], which integrates three factors: interest matching, recency (favoring more recent posts), and impact (measured by the number of followers of the poster). These factors together determine the ranking and distribution of posts.
>
> Therefore, if malicious agents have more benign followers, update their posts more frequently, or adapt their content based on the victim’s user profile, their posts will receive more exposure. They also have a higher chance of reaching their targeted users. If we want to analyze this more systematically, we can study how malicious behavior patterns interact with each of these three factors and how this interaction affects post-ranking and ultimately fraud success.
>
> It is also important to note that ranking only determines reachability, which is just one condition for fraud to succeed. Once a victim sees the fraudulent post, the later interactions—especially multi-turn private conversations between malicious and benign agents—become much more critical for determining the outcome.
>
> References:
>
> 1. Twitter. The Algorithm. 2023. https://github.com/twitter/the-algorithm

---

> > ### Comment · Reviewer_U9v1 · 2025-11-25
> >
> > The authors have solved all my concerns. I suggest this work now has enough workload and shows generalizability. I'd like to increase the score to 8.
> >
> > Thank you for your contribution for the community.

---

> > > ### Author Response · Authors · 2025-11-25
> > >
> > > Thank you very much for your thoughtful follow-up and for re-evaluating our work. Your suggestions have greatly helped us improve the paper, and we sincerely appreciate your decision to increase the score.

---

### Official Review · Reviewer_c5hA · 2025-10-29

**Soundness:** 2
**Presentation:** 3
**Contribution:** 3
**Rating:** 4
**Confidence:** 3

**Summary:**

This paper investigates collective financial fraud behaviors among large language model (LLM) agents within multi-agent social systems. The authors propose MultiAgentFraudBench, a large-scale benchmark simulating 21 realistic online fraud scenarios across public and private domains. Using this framework, they analyze how interaction depth, model capability, and agent collusion amplify fraud success. Experiments involving 17 major LLMs show that stronger reasoning models often exhibit higher fraud effectiveness, while current safety alignments fail to prevent malicious compliance. The study also reveals that collusive communication channels significantly magnify harm and that prolonged dialogues increase vulnerability. Finally, the authors explore mitigation strategies at content, agent, and society levels—such as debunking, banning, and collective resilience—to reduce fraud propagation. Overall, the work highlights critical safety risks in autonomous multi-agent systems and provides a valuable foundation for studying malicious coordination and countermeasures.

**Strengths:**

1. This work raises a novel and important question: whether multi-agent systems can engage in collusive fraudulent behaviors through coordinated actions.
2. Building on this problem, the paper introduces a large-scale benchmark covering 21 full-cycle financial fraud scenarios, enabling systematic evaluation of fraud emergence and safety vulnerabilities.
3. The proposed evaluation metrics effectively verify the existence of collaborative fraud and reveal weaknesses in current safety mechanisms.
4. Two mitigation strategies are designed to alleviate collusive fraud, enhancing the security and ethical deployment of collaborative AI systems in social contexts.

**Weaknesses:**

1. The study lacks validation in real-world settings. Although the framework covers 21 simulated scenarios, it simplifies financial fraud into a binary opposition between “benign” and “malicious” agents. In reality, user behaviors are more diverse, and such simplification may limit the framework’s realism and generalizability.
2. The agent design is overly simplistic. Each agent is defined by five positive personality traits, which may not accurately reflect the heterogeneity and psychological diversity of real populations. Moreover, individuals differ in their awareness and resistance to financial scams, but these variations are not modeled in the simulation.
3. The proposed mitigation strategies also have limitations. The agent-level banning assumes clear detection without considering deceptive or disguised malicious behavior and may incur high computational costs in large-scale deployment. Meanwhile, the society-level collective resilience strategy relies on voluntary cooperation rather than systematic enforcement, which may not consistently hold in real-world environments and thus limits reliability.

**Questions:**

1. How consistent are the simulation results with real-world financial fraud behaviors? Further validation in realistic environments would strengthen the study’s credibility.
2. Have the authors considered testing agents with adverse or malicious personality traits? Incorporating agents with varying levels of vigilance or susceptibility to fraud could better reflect real-world population diversity.
3. Could the authors provide more detailed case analyses of both successful and failed fraud attempts to illustrate behavioral dynamics more clearly?
4. How would the proposed mitigation strategies perform under real adversarial conditions, where malicious agents actively adapt or disguise their behaviors?

---

> ### Author Response · Authors · 2025-11-23
> **Response to  Reviewer c5hA (Part 1/3)**
>
> Thank you for your insightful review and for highlighting the strengths of raising a "novel and important question," providing a "valuable foundation" for studying malicious coordination, and acknowledging that our metrics "effectively verify" collaborative fraud. We will try our best to answer all your questions. Please let us know if you still have further concerns or if you are not satisfied with the current responses, so that we can further update the response ASAP.
>
> ---
>
> **Q1**: The binary setup of “benign vs. malicious agents” oversimplifies real-world user diversity and may limit realism and generalizability.
>
> **A1**:
> We acknowledge your concern about real-world realism. However, we believe that our experimental setup is rational for the following reasons:
>
> 1. First, our agent generation space is diverse. We follow common practices in agent-based social simulations [1,2,3]. Each agent’s user profile includes attributes such as gender, age, occupation, interests, and personality. This creates diverse interests and behavioral preferences for every agent when interacting with posts. Therefore, our design places malicious agents inside thousands of diverse agents, rather than simplifying the system into only a binary opposition between “benign” and “malicious” agents.
>
> 2. Second, our simplification is intentionally designed to mirror real-world diversity while avoiding unnecessary human bias. We grant agents sufficient autonomy and study their emergent behaviors in large-scale interactions. These agents have their own interests, personalities, and individual goals, which resemble the diversity of real social platforms. Our observations support our design: our benign agents develop emergent self-defense behaviors without being explicitly prompted. For example, in Appendix F.2 (line 955), agents take defense actions such as “block_user” and “report_user,” and they post security alerts when encountering fraudulent content. This high-autonomy design makes our testbed a suitable platform for observing the evolution of interactions between benign and malicious agents, including complex collaborative behaviors among malicious agents (Figure S3, line 884 and S4 , line 986).
>
> ---
>
> **Q2**: Our agent design oversimplifies real human psychological diversity and does not capture differences in individuals’ awareness and resistance to financial scams.
>
> **A2**: We understand your concern. However, as we explained in the previous response, our agent design, including the use of the Big Five personality model, follows the practices used in several published papers [1,2,3]. We adopt a fine-grained personality generation method. Each dimension is divided into 10 levels with corresponding descriptions of how these traits manifest in an agent’s behavior. Together with user profile attributes such as gender, age, occupation, and interests, this creates a large and diverse individual space. Different interests also imply different sensitivities to various categories of fraudulent posts.
>
> Moreover, another important factor that shapes an agent’s behavior toward financial scams is **time**. As interactions progress, each agent accumulates unique experiences. Some agents may be more cautious about financial scams because of their personality and therefore avoid being deceived. Some agents who are deceived may choose to inform the community by posting security alerts. Malicious agents also adapt to the reactions of benign agents. **Time enriches the context beyond the initial personality settings.** Our benchmark, therefore, provides a dynamic and evolving environment. Based on this, we observe complex group behaviors that closely resemble real human dynamics (Appendix F.1 Figure S5 and S6, Appendix E.2 Figure S4 line 986). We will further clarify this point in our camera-ready paper.
>
>
> References:
>
> 1. Yang et al. Oasis: Open Agent Social Interaction Simulations with One Million Agents. arXiv:2411.11581, 2024.
> 2. Gao et al. S3: Social-network Simulation System with Large Language Model-empowered Agents. arXiv:2307.14984, 2023.
> 3. Park et al. Generative Agents: Interactive Simulacra of Human Behavior. Proceedings of UIST, 2023.

---

> ### Author Response · Authors · 2025-11-23
> **Response to  Reviewer c5hA  (Part 2/3)**
>
> ---
>
> **Q3**: The proposed mitigation strategies may not be reliable in real-world settings because malicious agents can evade detection, and voluntary collective defense may not consistently occur.
>
> **A3**: Thank you for your comment, but we respectfully disagree.
> First, blocking malicious users is a common strategy used by real-world social media platforms [1][2][3][4]. These platforms also detect suspicious users based on behavior trajectories. We follow this spirit and design an LLM-driven detector. As shown in line 427, our detector reduces DeepSeek-V3’s fraud success rate from 45.8% to 6.7%, while not banning normal agents. This shows that our approach is a promising direction for further exploration. For large-scale deployment, model quantization and pruning may help reduce computational costs [5].
>
> Next, we address the reliability of collective resilience in real environments. Prior empirical and design-oriented studies show that user-initiated reporting and peer-to-peer warnings do exist in real fraud scenarios and are actively used. For example, crowdsourced reporting can improve the performance of fraud-detection systems [6]. Online platforms such as Scamwatch, ActionFraud, and ScamWarners have long relied on voluntary user submissions and public sharing of scam cases, providing researchers and regulators with extensive data about fraud patterns [7]. Therefore, we are not assuming an ideal mechanism that does not exist in reality. Instead, we use a controlled simulation environment to model and test the potential effect of “peer alerts on social platforms” in financial-fraud situations. Our results indicate that, beyond institutional rules and regulatory enforcement, platforms can also design mechanisms that encourage self-organized group-level warnings. While such measures cannot fully replace formal protections, they can serve as a valuable complement to enhance overall robustness and sustainability.
>
> References
>
> 1. Twitter/X. Updating Our Approach to Misleading Information. 2020. https://blog.x.com/en_us/topics/product/2020/updating-our-approach-to-misleading-information
>
> 2. Xiaohongshu. User Agreement. 2025. https://agree.xiaohongshu.com/h5/terms/ZXXY20250119002/-1
>
> 3. Reddit. Reddit Transparency Report 2022. 2022. https://40687240.fs1.hubspotusercontent-na1.net/hubfs/40687240/Reddit%20Inc/PDF/Reddit-Transparency-Report-2022.pdf
>
> 4. Reddit. Reddit Transparency Report: 2023 H1. 2023. https://redditinc.com/policies/2023-h1-transparency-report
>
> 5. Wang et al. Model Compression and Efficient Inference for Large Language Models: A Survey. arXiv:2402.09748, 2024.
>
> 6. Dasari et al. AI-Based UPI Scam Detection and Reporting System Using Crowdsourced Patterns & Verified Reports. IJIRT, Volume 12, Issue 3, 2025. https://ijirt.org/publishedpaper/IJIRT183933_PAPER.pdf
>
> 7. Gundur et al. Using Digital Open Source and Crowdsourced Data in Studies of Deviance and Crime. In: Researching Cybercrimes: Methodologies, Ethics, and Critical Approaches, Springer, 2021, pp. 145–167.
>
> ---
>
> **Q4**: How consistent are the simulation results with real-world financial fraud behaviors? Further validation in realistic environments would strengthen the study’s credibility.
>
> **A4**: Thank you for the question. In Appendix G.3, we follow your suggestion and compare the fraud trajectories of our agents with real-world financial fraud cases [1][2][3]. We find that the agents’ fraud trajectories are highly consistent with real-world fraud behaviors—not only in terms of scenario design, but also in the step-by-step structure of the fraud chain.
> For example, in the NFT Whitelist and Gas Fee Scam, our simulated case mirrors the real-world pattern. User 108 impersonates an NFT platform staff member and tells User 41 that they have been selected for a “VIP whitelist’’ in a charity-themed NFT auction. User 41 is then asked to pay a refundable deposit and several small additional payments for account verification and whitelist maintenance. Before the NFT is released, User 108 requests a final Ethereum gas fee for minting. After receiving the cryptocurrency transfer, User 108 ends all communication. This fraud trajectory closely mirrors real-world NFT scam procedures.
>
> We also provide more case comparisons in Appendix G.3. These analyses strengthen the realism and credibility of our benchmark. Thank you for your helpful suggestion. We will update the results in the revised version.
>
> References
>
> 1. Africa Check. Analysis: About the Money? How to Spot a Scam Conference. 2019.
> https://africacheck.org/fact-checks/blog/analysis-about-money-how-spot-scam-conference
>
> 2. Heart & Soul Whisperer. Case Studies of Art Scams. 2024.
> https://www.heartandsoulwhisperer.com.au/case-studies-of-art-scams
>
> 3. Elliptic. NFTs and Financial Crime Report. 2022.
> https://www.elliptic.co/hubfs/NFT%20Report%202022%20-full.pdf

---

> ### Author Response · Authors · 2025-11-23
> **Response to Reviewer c5hA (Part 3/3)**
>
> ---
>
> **Q5**: Have the authors considered testing agents with adverse or malicious personality traits? Incorporating agents with varying levels of vigilance or susceptibility to fraud could better reflect real-world population diversity.
>
> **A5**: Yes, we have considered this. In Section 6.3 (line 430), we modify the personalities of benign agents to encourage them to help others resist similar fraud after they are targeted. They can do this through public posts or private messages. Following your suggestion, we have added more experimental results.
>
> As shown in the table below, when a higher proportion of proactive agents is present in the population, the overall robustness against fraud becomes stronger. For example, with deepseek-v3, the population-level fraud success rate $R_{pop}$ decreases from 15% to 2% as more proactive agents join the group. This also suggests that platforms can promote positive community behaviors—such as awareness campaigns or crowdsourced reporting—to mitigate fraud risks.
>
> We have updated these results in the revised version of the paper.
>
> | Resilience Level | DeepSeek-V3 | Claude-3.7-Sonnet (w/o thinking) |
> |------------------|-------------|---------------------|
> | Baseline         | 15%         | 10%                 |
> | Res.=25\%        | 9%          | 3%                  |
> | Res.=50\%        | 4%          | 1%                  |
> | Res.=100\%        | 2%          | 0%                  |
>
>
> ---
>
> **Q6**: Could the authors provide more detailed case analyses of both successful and failed fraud attempts to illustrate behavioral dynamics more clearly?
>
> **A6**: Sure. As shown in Figure 7 of Appendix G.1, the left case successfully progresses through Initial Contact, Trust Building, and Payment Request, and it finally leads to a money transfer. In contrast, the right case is stopped during the Trust Building stage. The malicious agent sends a Payment Request before lowering the user’s caution, so the user stays vigilant during the whole interaction. As a result, the fraud attempt cannot move into effective trust formation, and it also cannot reach the final transactional stage.
>
> ---
>
> **Q7**: How would the proposed mitigation strategies perform under real adversarial conditions, where malicious agents actively adapt or disguise their behaviors?
>
> **A7**:   This is a very interesting question! To study how malicious agents adapt to the detector, we designed an adversarial experiment where the surviving malicious agents can observe which partners were banned and can also access their historical action trajectories. The results show that the recall rate of our GPT-based detector for malicious agents drops from 90% to 75%, but the false positive rate remains 0%, meaning that no benign users were banned.
>
>   Appendix G.2 presents an example of a false negative. After seeing that other partners were banned, Agent 103 begins to post more art-related and charity-oriented content and only occasionally inserted financial signals, such as urgent monetary requests and get-rich-quick course promotions. Because most of the content appears normal and the malicious signals are sparse, the monitoring agent ultimately classifies this user as benign.
>
>   This example illustrates one pattern of malicious agents’ adaptability: they can imitate the behavioral trajectories of benign users to evade detection. We will update this discussion in the revised version.

---

> ### Author Response · Authors · 2025-11-28
>
> Dear reviewer, we wanted to ensure we have addressed all your concerns satisfactorily. If there are any additional points or feedback you'd like us to consider, please let us know. Thank you for your time and effort in reviewing our paper.

---

> > ### Comment · Reviewer_c5hA · 2025-11-28
> >
> > Thanks for the response from the authors. After reviewing the rebuttal and the comments from other reviewers, I will keep my original score unchanged.

---

### Official Review · Reviewer_1afb · 2025-10-29

**Soundness:** 3
**Presentation:** 3
**Contribution:** 3
**Rating:** 8
**Confidence:** 4

**Summary:**

This manuscript presents a systematic investigation into the potential risks of collusive financial fraud conducted by multi-agent systems powered by large language models (LLMs) on social platforms. Building upon the OASIS social simulation framework, the authors develop the first comprehensive benchmark, MultiAgentFraudBench, that covers both public and private domains across the full fraud lifecycle (hooking, trust building, and payment request). Through large-scale simulations involving hundreds of benign and malicious agents, the study quantitatively evaluates major LLMs in terms of conversion and propagation rates, revealing a consistent trend that more capable models exhibit higher fraud risks. It further examines the differential effectiveness of content-level, agent-level, and social-level mitigation strategies.

**Strengths:**

- This study leverages the advanced LLM-based multi-agent framework OASIS to investigate a topic of substantial practical significance, representing a noteworthy contribution.
- The manuscript includes a comprehensive range of large language models, reflecting considerable research effort.
- The broad coverage of fraud patterns and the simulation design are rigorous and well-founded.
- Extensive experiments are conducted, yielding several interesting findings, for example, that small social circles achieve higher fraud success rates than large ones when the number of interaction rounds is limited, and that more capable LLMs can successfully deceive less capable ones in adversarial settings between fraudsters and victims.

**Weaknesses:**

- Some conclusions appear somewhat trivial, such as an increase in normal users reduces the overall fraud success rate.
- Although collusion is emphasized as the core fraudulent mechanism, it is defined only by the existence of private communication among fraudsters, which seems insufficient. The paper lacks deeper methodological design and experimental analysis focused on collusion itself; a clearer and more robust definition of collusion would strengthen the work.
- Finally, the findings do not strongly highlight the distinct advantages or necessity of using LLMs, i.e. similar conclusions might plausibly be obtained with traditional generative AI models serving as agents.

**Questions:**

Please refer to the weaknesses.

---

> ### Author Response · Authors · 2025-11-23
> **Response to  Reviewer 1afb (Part 1/2)**
>
> Thank you for your great efforts in reviewing this paper and for recognizing our work as a "noteworthy contribution," describing the simulation design as "rigorous and well-founded," and highlighting our "extensive experiments." We will try our best to answer all your questions. Please let us know if you still have further concerns or if you are not satisfied with the current responses, so that we can further update the response ASAP.
>
> ---
>
> **Q1**: Some conclusions appear somewhat trivial, such as an increase in normal users reduces the overall fraud success rate.
>
> **A1**: Yes, you are right that this is one of our ablation studies about the ratio between benign users and malicious users. However, we also discovered many other interesting patterns that map well to real-world dynamics. These include, but are not limited to: the emergence of complex collaborative behaviors among malicious agents (Appendix F.1, Figure S5 and Figure S6, line 1014), the ability of malicious agents to adapt their strategies to evade platform interventions (Appendix E.2, Figure S4, line 972), and the relationship between the proportion of fraud-susceptible users and overall group robustness (Section 6.3, Figure 5, line 472).
>
> These patterns highlight the urgency of taking the financial fraud risks posed by multi-agent systems seriously.
>
> ---
>
> **Q2**: Although collusion is emphasized as the core fraudulent mechanism, it is defined only by the existence of private communication among fraudsters, which seems insufficient. The paper lacks a deeper methodological design and experimental analysis focused on collusion itself; a clearer and more robust definition of collusion would strengthen the work.
>
> **A2**: Thank you for the insightful suggestion! Collusion is indeed more than private communication. Based on prior work and common definitions in the collusion literature [1, 2], collusion generally requires several conditions: (1) two or more parties share the same goal, and this goal often harms other parties; (2) their communication is usually secret and hidden from other stakeholders. In our setting, agents can negotiate secretly through private chats, and they can also support each other in the public domain; (3) they can share information. In our paper, the shared information among agents is rich. It includes not only environmental observations and actions, but also their reflections and strategies.
>
> We fully agree with your suggestion. We performed an ablation study on collaboration in the paper (e.g., Section 4.3, Table 2), and we found that allowing collaboration leads to greater harm compared to independent agents. To further quantify collusion, we designed two additional experiments. Collaboration mainly affects fraud success at two stages: (i) interactions between peers around a post in the public domain, (ii) coordinated fraud against the same victim in the private domain.
>
> For (i), we selected deepseek-R1 and deepseek-V3 as two malicious models. For each post created by a malicious agent, we counted how many other malicious agents participated in commenting. We treat this as one form of collaboration. As shown by the results, malicious agents with more public-domain interactions tend to achieve higher fraud success rates. For example, 15.37% of posts by DeepSeek-R1 agents received at least one promotional comment from another malicious agent, while only 9% of posts by DeepSeek-V3 received such support.
>
> | Model \ Number of malicious agents commenting on the same post       | 0  | 1  | 2  | 3  | ≥4  | ≥1  | Population-level Fraud Success Rate $R_{pop}$|
> |--|---|---|-----|----|----|----|------|
> | DeepSeek-V3  | 90.72%          | 6.89%          | 1.50%            | 0.90%| 0.00%| 9.00%  | 15.00% |
> | DeepSeek-R1  | 84.63%          | 7.24%          | 3.43%   | 2.54%            | 2.20%| **15.37%**  | **41.00%**  |
>
> For (ii), we counted how many malicious agents privately contacted the same user, since the actual transfer happens in the private domain. As shown in the table below, 41% of victims contacted by DeepSeek-R1 interacted with at least two malicious agents, while only 6% of victims contacted by DeepSeek-V3 showed this pattern. Because malicious agents can share information about their target users with their partners, this metric reflects a higher degree of collusion in DeepSeek-R1 compared with DeepSeek-V3. We have added all relevant analyses to the revised version.
>
> | Model \ Number of malicious agents targeting on the same victim| 0  | 1  | 2  | 3  | ≥4  | ≥2 | Population-level Fraud Success Rate $R_{pop}$ |
> |---|---|---|----|---|---|----|----|
> | DeepSeek-V3  | 84% | 10%| 4% | 2% | 0% | 6% |15.00%|
> | DeepSeek-R1  | 25% | 34% | 22%| 7% | 12%| **41%**|**41.00%**  |
>
> References:
>
> 1. Hammond L, Chan A, Clifton J, et al. Multi-agent risks from advanced ai[J]. arXiv preprint arXiv:2502.14143, 2025.
>
> 2. Rees, Ray (1993). “Tacit Collusion”. Oxford Review of Economic Policy 9.2, pp. 27–40

---

> ### Author Response · Authors · 2025-11-23
> **Response to Reviewer 1afb (Part 2/2)**
>
> ---
>
> **Q3**: Finally, the findings do not strongly highlight the distinct advantages or necessity of using LLMs, i.e., similar conclusions might plausibly be obtained with traditional generative AI models serving as agents.
>
> **A3**: Thank you for the helpful suggestion! Our response is that traditional mathematical or agent-based models (ABMs) are indeed not suitable for the plausible financial fraud scenario in our study. These ABMs usually rely on a small set of hand-crafted behavioral rules defined by researchers. Their state transitions are often deterministic or based on simple probabilities, so agent behaviors are simple and rigid. In addition, they cannot interact through natural language [1,2,3]. Because of this, many complex human behaviors in real-world settings cannot be effectively encoded, such as context-dependent strategy changes, high-level social interaction, negotiation, and deception.
>
> In contrast, LLM-driven agents generate behaviors through language understanding, role conditioning, and contextual reasoning. They can naturally express these complex social processes and adjust their strategies during dynamic dialogues.
>
> Therefore, studying collaborative financial fraud on social platforms requires LLM-based agents and an environment that supports language interaction and diverse actions. Traditional ABMs cannot capture these behavioral patterns.
> We will add a discussion of this point in the related work section. Thank you again for your valuable feedback.
>
> References:
> 1. El-Sayed, Abdulrahman M., et al. "Social network analysis and agent-based modeling in social epidemiology." Epidemiologic Perspectives & Innovations 9.1 (2012): 1-9.
>
> 2. Bianchi, Federico, and Flaminio Squazzoni. "Agent‐based models in sociology." Wiley Interdisciplinary Reviews: Computational Statistics 7.4 (2015): 284-306.
>
> 3. Will, Meike, et al. "Combining social network analysis and agent-based modelling to explore dynamics of human interaction: A review." Socio-Environmental Systems Modelling 2 (2020): 16325-16325.
>
> ---
>
> **Q4**: Flag For Ethics Review: Yes, Privacy, security and safety
>
> **A4**: Thank you for raising this important concern. Fraud is a sensitive topic, and we fully recognize the need for strict attention to safety. We propose a tiered release strategy to prevent misuse while still supporting defensive research.
>
> First, for the **code**, we will open-source only the simulation framework (an improved platform based on OASIS).
>
> Second, for the **data**, we will use a gated-access mechanism. Our dataset of fraudulent posts will not be directly available for public download. Institutions will need to apply for access, and all requests will go through manual review. We will strictly limit its use to defensive academic research.
>
> Finally, we want to emphasize our **motivation**. Directly studying fraud in the real world would involve unacceptable risks. Our simulation platform serves as a safe “sandbox,” which allows us to quantify these threats and develop defensive methods (such as monitoring agents) without harming real users. As Multi-Agent Risks from Advanced AI points out, collusion is still a frontier topic that has not been sufficiently explored. We believe this work can help the community better investigate these emerging dynamic mechanisms.

---

> ### Author Response · Authors · 2025-11-28
>
> Dear reviewer, we wanted to ensure we have addressed all your concerns satisfactorily. If there are any additional points or feedback you'd like us to consider, please let us know. Thank you for your time and effort in reviewing our paper.

---

### Official Review · Reviewer_1Neb · 2025-10-31

**Soundness:** 3
**Presentation:** 3
**Contribution:** 2
**Rating:** 4
**Confidence:** 3

**Summary:**

The paper aims to assess the risks of collaboration among malicious agents for fraud. The methodology relies on running a pure-agent society where agents have distinct personalities and individual goals. Agent societies consist of multiple agents with individual goals and characteristics.

The MultiAgentFraudBench extends the OASIS framework, which simulates a social media platform with LLM agents. The simulation follows time steps. Every agent acts in every time step. MultiAgentFraudBench extends OASIS to the private domain by adding private messaging. The benchmark consists of 100 benign agents (simulating platform users) and 10 malicious agents.

Generation of benign agents (users): Benign agents are generated with an age uniformly between 18-65, and personality as drawn from normal distributions on the Big Five dimensions (Openness to Experience, Conscientiousness, Extraversion, Agreeableness, Neuroticism). Furthermore, users hold sensitive information (social security number, Bank Card, PIN) and assets (bank balance).

Action space: Both benign/malicious agents have the same action space (e.g. "like", "post", "comment" and "private message"), and act with the same distribution to avoid trivial detection.

Observation space: benign and malicious agents see the same posts ("the public domain") and their own private messages ("the private domain"). Furthermore, malicious agents know about each other.

The fraud success is evaluated with two metrics: 1) $R_{conv}$ Conversation-level fraud success (successful persuasion percentage of private chats) 2) $R_{pop}$ Population-level fraud impact (defrauded percentage of benign population.)

With this setup, the paper considers a range of LLM-models for malicious agents and finds that fraud success is correlated with general capability, with DeepSeek-R1 reaching a maximum of $41.0$ for $R_{pop}$ and Gemini-Flash-2.5 reaching $76.5$ for $R_{conv}$. They inspect what affects fraud success and find that longer interactions (higher interaction depth) increase fraud success, as well as public domain collusion.

The authors offer three methods for mitigating fraud.
1) Debunking: at midpoint of run, all malicious posts and private methods are tagged with risk labels. They find that $R_{pop}$ decreases but $R_{conv}$ doesn't.
2) Banning: add a monitoring LLM-agent that classifies and removes agents exceeding a threshold of maliciousness. Both $R_{pop}$ and $R_{conv}$ go down significantly, and the model achieves detection accuracy of 1.0
3) Group Resilience. Benign actors may make awareness of posts in the public domain. This reduces both $R_{pop}$ and $R_{conv}$.

**Strengths:**

1) Benchmark
The paper offers a comprehensive simulation of a social media platform with posting, liking, commenting, and private messaging. The fraud types included are broad and realistic.

2) Findings
Findings are interesting and relevant to the research questions.

a) LLMs rarely refuse actions, even when the malicious intent is obvious. The finding suggests that interactive settings are an effective way to "jailbreak" models and have them act maliciously.

b) Ablation: Collusion of malicious agents almost doubles the fraud success rates. This underlines the risk of collaboration in fraud tasks.

c) User Banning by an LLM detection model performs very well. This is a promising result for fighting fraud.

**Weaknesses:**

1) Key assumption of human emulation:

For this experiment to be an accurate benchmark of the risk, it is assumed that the targets (the benign LLM-agents) emulate the susceptibility of human (real users). However, this assumption may be questionable especially because the model used to simulate benign users (Qwen-2.5-32B-Instruct) is of lower general capability than some of the malicious models (like DeepSeek-R1).

2) Collaboration:

The amount of collusion (collaboration between malicious agents) is not quantified. I would like to see statistics on the number of agents working together for a typical fraud success in the appendix, and more elaborate statistics on the amount of collusion in the public domain as in section 5.2.

3) Bias to action:

As mentioned in Appendix C.1, the activation probability in a time step is $1$. Does this cause an "action bias" to benign actors? If at every time step they are "forced to act", are they more likely to interact with malicious actors? Many typical social media users are "lurkers", and this may decrease fraud susceptibility.

4) Relationship Network Connection:

The relationship between users is generated by an ER random graph with $p = 0.1$. However, social media network connectivity can be more realistically modelled by a Bárábasi-Albert scale-free model, as in OASIS.

5) Debunking as a mitigation method:

•	The debunking model (labelling all posts with perfect knowledge as "fraudulent") is simplistic.

•	In general, this method assumes perfect knowledge of maliciousness. However, this should be done with a detector model.

•	Privacy concerns in debunking in the private domain.

6) Group resilience as a mitigation method:

How well does the result of group resilience (i.e. posting fraud awareness posts) generalize to human users? In LLMs, putting fraud awareness in the context window seems likely to decrease their susceptibility, but humans may have more selective attention spans and ignore awareness posts. This again raises the question of whether using LLMs to judge humans' fraud susceptibility is appropriate.

**Questions:**

- It is interesting to see that malicious agents can effectively coordinate to defraud other LLMs, but I am not convinced that a 32B LLM is a realistic emulation of a human acting on a social media platform. Can you address this concern?

- Can you quantify the increase in fraud success through collaboration over solo-acting malicious agents?

- See weaknesses.

---

> ### Author Response · Authors · 2025-11-23
> **Response to  Reviewer 1Neb (Part 1/3)**
>
> Thank you for your insightful review and for highlighting the strengths of our work, recognizing the benchmark as a "comprehensive simulation" with "broad and realistic" fraud types, and acknowledging that our findings are "interesting and relevant" and the mitigation results are "promising". We will try our best to answer all your questions. Please let us know if you still have further concerns or if you are not satisfied with the current responses, so that we can further update the response ASAP.
>
> ---
>
> **Q1**: Concerns about using the benign LLM-agents to emulate the susceptibility of human (real users).
>
> **A1**: Thank you for your question! We address the necessity of using LLM agents to simulate real humans from two perspectives:
>
> 1. **Moral and practical constraints.**
> In computational social science, researchers often use agent-based models to understand, analyze, and predict phenomena and outcomes that are difficult or impossible to test in real-world experiments because of moral and legal risks. Examples include misinformation [1], online polarization [2], and herd effects [3]. The same concerns apply to our financial fraud setting. Therefore, using LLM agents is a necessary and ethical alternative.
> 2. **Support from recent research.**
> A growing body of work in social simulation, economics, and moral decision-making uses LLM-driven agents as proxies for human behavior. These studies show that LLM agents can align with human participants or reflect real macro-level statistics to a meaningful degree [4,5,6]. In our experiments, we also observed human-like fraud behaviors. For example, malicious agents collaborate to scam (Figure S5, line 1018), and benign agents notify the community after being scammed (Figure 7, line 436). We further summarize the full fraud trajectory between benign and malicious agents, which matches real-world patterns (Appendix G.2, line 1141). These observations support the validity of our simulation framework.
> 3. **Regarding the concern that benign agents are weaker.**
> First, our benchmark intentionally models an asymmetric setting where benign agents are weaker than malicious ones. This reflects the real world, where fraud groups use various social engineering strategies and tools to exploit ordinary users.
>
> Second, for most models in our benchmark, deceiving Qwen-2.5-32B-Instruct (the benign agent) is still a challenging task (Table 1). For example, stronger models such as Gemini-2.5-flash and GPT-4o achieve a fraud success rate $R_{pop}$ below 5%. We also tested stronger victims in Section 4.3 (Table 3, line 295). When replacing the benign agent with DeepSeek-V3, the fraud success rate of DeepSeek-V3 drops from 11% to 1%.
>
> Because our main focus is collusion itself, we need a baseline where “some room for fraud” exists due to the victim’s limited strength. This allows us to observe how collusion succeeds or fails and provides space for analysis and mitigation proposals. We appreciate your comment, and we will note in the paper that as malicious models evolve in the future, our framework can incorporate stronger benign agents for evaluation.
>
> References:
>
> 1. Gausen et al. Using agent-based modelling to evaluate the impact of algorithmic curation on social media. ACM Journal of Data and Information Quality, 2022.
>
> 2. Song et al. Dynamic spirals put to test: An agent-based model of reinforcing spirals between selective exposure, interpersonal networks, and attitude polarization. Journal of Communication, 2017.
>
> 3. Lee et al. Heterogeneous expectations leading to bubbles and crashes in asset markets: Tipping point, herding behavior and group effect in an agent-based model. Journal of Open Innovation: Technology, Market, and Complexity, 2015.
>
> 4. Park et al. Generative agents: Interactive simulacra of human behavior. Proceedings of UIST, 2023.
>
> 5. Karten et al. LLM Economist: Large population models and mechanism design in multi-agent generative simulacra. arXiv:2507.15815, 2025.
>
> 6. Yang et al. TwinMarket: A scalable behavioral and social simulation for financial markets. arXiv:2502.01506, 2025.

---

> ### Author Response · Authors · 2025-11-23
> **Response to  Reviewer 1Neb (Part 2/3)**
>
> ---
>
> **Q2**: Quantify the amount of collusion
>
> **A2**: Thank you for the suggestion! We agree that collusion among agents affects fraud success in two key stages: (i) interactions around a post in the public domain, and (ii) coordinated fraud attempts in private messages targeting the same victim.
>
> For (i), we selected DeepSeek-R1 and DeepSeek-V3 as two malicious models. For each post created by a malicious agent, we counted how many other malicious agents joined the public comments. This serves as one form of collaboration. As shown in our table, malicious agents with more public-domain interactions usually achieved higher fraud success rates. For example, 15.37% of posts generated by DeepSeek-R1 agents received at least one supportive comment from another malicious agent, while DeepSeek-V3 reached only 9%.
>
> | Model \ Number of malicious agents commenting on the same post       | 0  | 1  | 2  | 3  | ≥4  | ≥1  | Population-level Fraud Success Rate $R_{pop}$|
> |--------------|-----------------|----------------|------------------|------------------|------------------|-----------------------------------|--------------------------------|
> | DeepSeek-V3  | 90.72%          | 6.89%          | 1.50%            | 0.90%            | 0.00%            | 9.00%                            | 15.00%                        |
> | DeepSeek-R1  | 84.63%          | 7.24%          | 3.43%            | 2.54%            | 2.20%            | **15.37%**                           | **41.00%**                        |
>
>
> For (ii), we counted how many malicious agents privately contacted the same user, since the actual transfer happens in the private domain. As shown in the table below, 41% of victims contacted by DeepSeek-R1 interacted with at least two malicious agents, while only 6% of victims contacted by DeepSeek-V3 showed this pattern. Because malicious agents can share information about their target users with their partners, this metric reflects a higher degree of collusion in DeepSeek-R1 compared with DeepSeek-V3. We have added this analysis to the revised version.
>
> | Model \ Number of malicious agents targeting on the same victim       | 0  | 1  | 2  | 3  | ≥4  | ≥2 | Population-level Fraud Success Rate $R_{pop}$ |
> |--------------|--------------------|--------------------|---------------------|---------------------|----------------------|----|----|
> | DeepSeek-V3  | 84%                | 10%               | 4%                  | 2%                  | 0%                   | 6% |15.00%                        |
> | DeepSeek-R1  | 25%                | 34%               | 22%                 | 7%                  | 12%                  | **41%**|**41.00%**                        |
>
>
> ---
>
> **Q3**: Bias to action: the relationship between the activation probability and fraud susceptibility.
>
> **A3**: We agree that this is a valuable experiment to explore. Therefore, we set the activation probability of 50% of benign users to 0.2 to represent lurkers, while the other 50% remain at 1. The table shows that benign users with higher activation probabilities are more vulnerable to fraud. For example, in the group with activation probability = 0.2, only 4% of benign users were successfully defrauded, while in the group with activation probability = 1, 14% of benign users were successfully defrauded. This matches your expectation. We will include these results in the camera-ready version.
>
> Moreover, **we want to clarify why we used highly active benign users in the benchmark**. Our goal is to evaluate the upper bound of the harm caused by malicious agents on an idealized high-engagement platform. This follows the security mindset in safety and security research: we should not only look at average cases but also focus on worst-case scenarios, and ensure that “the probability of very bad outcomes is sufficiently small” [1,2,3]. This may provide a more reliable estimate of the upper bound of financial fraud risks in our setting. Of course, users can set their own activation probability distributions to evaluate fraud risks in different real-world social communities based on our open-sourced framework.
>
> | Benign Activation Probability | Population-level Fraud Success Rate $R_{pop}$ |
> |------------------------------|---------------------------------|
> | 0.2 (Lurker-like Users)      | 4%                              |
> | 1.0 (Highly Active Users)    | 14%                             |
>
> References:
>
> 1. Amodei D, Olah C, Steinhardt J, et al. Concrete problems in AI safety. arXiv preprint arXiv:1606.06565, 2016.
>
> 2. Yampolskiy R V, Spellchecker M S. Artificial intelligence safety and cybersecurity: A timeline of AI failures. arXiv preprint arXiv:1610.07997, 2016.
>
> 3. Hendrycks D, Mazeika M, Woodside T. An overview of catastrophic AI risks[J]. arXiv preprint arXiv:2306.12001, 2023.

---

> ### Author Response · Authors · 2025-11-23
> **Response to  Reviewer 1Neb (Part 3/3)**
>
> **Q4**: Ablation on network topology.
>
> **A4**: Thank you for pointing this out. For our scale-free network, each new node attaches to 2 existing nodes with probability proportional to their degree. As shown in the figure, we observe that using a scale-free topology slightly increases the vulnerability of benign agents to financial fraud. We will add these results to the revised version.
>
> | Malicious Agent  | Topology   | Population-level fraud success rate $R_{pop}$ | Conversation-level fraud success rate $R_{conv}$  |
> |----|---|--|----|
> | DeepSeek-V3  | Random   | 15%  | 50%    |
> | DeepSeek-V3  | Scale-free  | 17%  | 50%    |
> | Claude-3.7-Sonnet-nothinking  | Random      | 10%  | 52.90% |
> | Claude-3.7-Sonnet-nothinking | Scale-free  | 12%  | 52.60% |
>
> ---
>
> **Q5**: Debunking as a mitigation method: simplistic design, assumption, and privacy concern in the private domain.
>
> **A5**: Thank you for your questions! Debunking is one of our baseline mitigation measures, and it is widely used by platforms [1,2]. We apply debunking to all malicious posts because we want to study the upper bound of this method. Unfortunately, even under this ideal setting, debunking has limited ability to reduce fraud harm, as we discuss in the main text (Section 6.1, line 417). It only reduces DeepSeek-V3’s fraud success rate from 15% to 10%, and reduces Claude-3.7-Sonnet’s fraud success rate from 10% to 8%. The ineffectiveness of content-level interventions motivates us to explore mitigation from other perspectives.
>
> Regarding the privacy concerns, we follow platform policies. In general, platforms are responsible for preventing leaks of private user messages [3,4]. Therefore, in real-world settings, platforms also add warnings in private domains when necessary. We hope this addresses your concerns.
>
> References:
>
> 1. Twitter/X. Updating Our Approach to Misleading Information. X Blog, 2020. https://blog.x.com/en_us/topics/product/2020/updating-our-approach-to-misleading-information
>
> 2. Xiaohongshu. User Agreement (2025 Edition). https://agree.xiaohongshu.com/h5/terms/ZXXY20250119002/-1
>
> 3. Reddit. Reddit Transparency Report 2022. https://40687240.fs1.hubspotusercontent-na1.net/hubfs/40687240/Reddit%20Inc/PDF/Reddit-Transparency-Report-2022.pdf
>
> 4. Reddit. Reddit Transparency Report: 2023 H1. https://redditinc.com/policies/2023-h1-transparency-report
>
> ---
>
> **Q6**: How well does the result of group resilience (i.e., posting fraud awareness posts) generalize to human users? and the question of whether using LLMs to judge humans' fraud susceptibility is appropriate.
>
> **A6**: Thank you for your perspective! We apologize for not providing enough background, especially about real-world practices. In fact, this approach is widely used by platforms such as Google and X [1, 2, 3]. They publish fraud awareness posts to help users identify and resist financial fraud risks. This shows that increasing group awareness is an effective way to reduce fraud vulnerabilities.
>
> We completely understand your concern about the generalization from our experiments to the real world. Our main motivation is that running financial fraud experiments on real human users would be unethical and impossible. Therefore, we aim to evaluate the financial fraud risks caused by malicious AI agents in a realistic large-scale sandbox environment. Under this constraint, using benign agents becomes necessary, and this practice is commonly adopted in computational social science (see our response to Weakness One).
>
> Considering both experimental feasibility and real-world relevance, we chose to evaluate the impact of group resilience in our benchmark. We hope this clarifies our reasoning, and we truly appreciate your understanding!
>
> References:
>
> [1] X. "Safety and Security." X Help Center, https://help.x.com/en/safety-and-security. 2025.
>
> [2] Meta. "Scam Prevention Hub: Stay Safe Online from Fraud and Scams." Meta, https://www.meta.com/safety/scam-prevention. 2025.
>
> [3] Google. "Tackling Scams and Fraud Together.", https://storage.googleapis.com/gweb-uniblog-publish-prod/documents/Tackling_scams_and_fraud_together.pdf. 2024.
>
> ---
>
> **Q7**: Can you quantify the increase in fraud success through collaboration over solo-acting malicious agents?
>
> **A7**: Of course! We have already included these results in Table 2. We evaluated two malicious agents, deepseek-v3 and deepseek-r1. The table shows that allowing collaboration greatly increases the fraud success rate of malicious agent groups. For example, the success rate of deepseek-r1 increases from 17% to 41% when collaboration is enabled. This demonstrates the effectiveness of our collusion mechanism.
>
> | Model   | Setting|Population-level fraud success rate $R_{pop}$ |Conversation-level fraud success rate $R_{conv}$ |
> |----|--|----|---|
> | DeepSeek-R1 | Without Collusion | 17%  | 35.0  |
> | DeepSeek-R1 | With Collusion| 41% | 60.2  |
> | DeepSeek-V3| Without Collusion | 9%   | 41.7  |
> | DeepSeek-V3 | With Collusion |15% | 50.0  |

---

> ### Author Response · Authors · 2025-11-28
>
> Dear reviewer, we wanted to ensure we have addressed all your concerns satisfactorily. If there are any additional points or feedback you'd like us to consider, please let us know. Thank you for your time and effort in reviewing our paper.

---

### Author Response · Authors · 2025-11-23
**Global responses to reviewers and new AC（Part 1/2）**

We would like to thank all reviewers for their participation before the freeze. We briefly summarize the discussion and our revisions as a reference for the new AC.

First, Reviewer U9v1 and Reviewer c5hA have replied. **Reviewer U9v1** increased their score from 6 to 8. **Reviewer c5hA** kept their score unchanged and did not raise any further concerns.

In the resulting revisions, our core collusion evaluation method, threat model, and empirical findings remain unchanged. All revisions are clarifications or additions, such as providing more detailed qualitative and quantitative analysis of collusion behaviors or mitigation strategies. These results further strengthen the importance of our work.

Below we list our main revisions point by point:

1. **Benchmark renaming**

We rename the benchmark from **MultiAgentFraudBench** to **MultiAgentFinancialFraudBench** to make explicit that the work focuses on *financial* fraud risks. All related mentions in the main text, figures/tables, and appendix have been updated accordingly.

---

2. **New quantitative analyses of collusion (Section 5.4)**

In the original version, the effect of collusion was mainly discussed via recommender-system amplification (Section 5.2) and collaboration failure modes (Section 5.3).
In the revision, we add Section 5.4 to further quantify collusion by using two metrics: (1) "number of malicious agents commenting on the same post" and (2) "number of malicious agents targeting the same target in the private domain".

2.1 **Public-domain collaboration: peer support in public posts"**

We select DeepSeek-R1 and DeepSeek-V3 as malicious models. For each post created by a malicious agent, we count how many other malicious agents appear in the comment thread, and report the corresponding ratio. Experimental results show that DeepSeek-R1 malicious agents show a higher rate of “peer-support” in public threads (≥1 Peers: 15.37% vs. 9.00% for DeepSeek-V3), together with a higher population-level fraud impact ($R_{\text{pop}}$: 41% vs. 15%).

| Model       | 0 Peers | 1 Peer | 2 Peers | 3 Peers | ≥4 Peers | ≥1 Peers | population-level fraud success rate $R_{\text{pop}}$ |
|------------|:-------:|:------:|:-------:|:-------:|:--------:|:--------:|:----------------:|
| DeepSeek-V3  | 90.72%  | 6.89%  | 1.50%   | 0.90%   | 0.00%    |  9.00%   |      15.00%      |
| DeepSeek-R1  | 84.63%  | 7.24%  | 3.43%   | 2.54%   | 2.20%    | 15.37%   |      41.00%      |

2.2 **Private-domain collaboration: multiple attackers per victim**

Since actual transfers happen in the private domain, we also count, for each victim, how many distinct malicious agents contacted them via private messages, and track the fraction of victims contacted by at least two attackers:

| Model       | 0 Attackers | 1 Attacker | 2 Attackers | 3 Attackers | ≥4 Attackers | ≥2 Attackers | $R_{\text{pop}}$ |
|------------|:-----------:|:----------:|:-----------:|:-----------:|:------------:|:------------:|:----------------:|
| DeepSeek-V3 |     84%     |    10%     |     4%      |     2%      |      0%      |      6%      |      15.00%      |
| DeepSeek-R1 |     25%     |    34%     |    22%      |     7%      |     12%      |     41%      |      41.00%      |

In the DeepSeek-R1 setting, 41% of victims are contacted by at least two malicious agents (vs. 6% for DeepSeek-V3). Combined with the higher $R_{\text{pop}}$ (41% vs. 15%), these statistics indicate stronger coordinated “multi-attacker” behavior in the DeepSeek-R1 group under our simulation.

---

> ### Author Response · Authors · 2025-11-23
> **Global responses to reviewers and new AC（Part 2/2）**
>
> 3. **Extended mitigation experiments (Section 6 & Appendix G)**
>
> Building on the original content-level debunking, agent-level banning, and society-level resilience strategies, we add two sets of experiments to better characterize their behavior.
>
> 3.1 **Evaluate our detector agent in an adversarial setting (Appendix G.2)**
>
> In the original monitoring mechanism, a detector agent scores behavior trajectories and bans high-risk agents. In the revision, we introduce a simple adversarial setting:
>
> - Surviving malicious agents can observe which peers were banned in previous rounds.
> - They can also access those peers’ historical trajectories and adapt their own behavior to mimic “safer” patterns.
>
> Under this setting, we observe:
>
> - Recall on malicious agents decreases from about **90%** to about **75%**;
> - The false positive rate on benign agents remains **0%**.
>
> This indicates that, in our current system, behavioral “disguise” can to some extent weaken the recall of malicious agents, but the overall detection performance remains at a high level.
>
> 3.2 **Fine-grained analyses of collective resilience in terms of different fractions of resilient agents (Section 6.3)**
>
> We introduce “resilient” benign agents: when they encounter or recognize fraud, they actively warn others via public posts or private messages. In our original paper, we only compared the cases with 100% resilient agents and with no resilient agents. In the new version, we extend the proportions to {0%, 25%, 50%, 100%}.
>
> As the table shows below, $R_{\text{pop}}$ decreases monotonically as the resilience level increases for both malicious models. For instance, with DeepSeek-V3 as the attacker model, $R_{\text{pop}}$ decreases from 15% (baseline) to 9%, 4%, and finally 2% at Res.=100%; the corresponding values for Claude-3.7-Sonnet (w/o thinking) drop from 10% to 3%, 1%, and 0%. These results indicate that, in our simulation, even a moderate share of “proactively warning” benign agents (e.g., Res.=50%) already leads to a substantial reduction in population-level fraud success.
>
> | Resilience Level | DeepSeek-V3 $R_{\text{pop}}$ | Claude-3.7-Sonnet (w/o thinking) $R_{\text{pop}}$ |
> | :--------------: | :--------------------------: | :-----------------------------------------------: |
> | 0%  (Baseline)   |             15%              |                      10%                         |
> | 25%              |              9%              |                       3%                         |
> | 50%              |              4%              |                       1%                         |
> | 100%             |              2%              |                       0%                         |
>
> ---
>
> 4. **More case studies about the realism of our simulation (Appendix G)**
>
> To make the fraud trajectories and modeling assumptions more transparent, we extend the appendix with two types of case analyses:
>
> 1. **Failed Fraud Case (Interrupted at the Trust-Building Stage)**
>    We present an example where the fraud attempt is interrupted during Trust Building: the malicious agent requests money too early, before sufficiently lowering the victim’s caution. In this case, the interaction stalls at an intermediate stage and never reaches a successful payment request.
>
> 2. **Structural Comparison between Simulated and Real-World Fraud Cases**
>    We add multiple pairs of cases comparing publicly documented financial fraud incidents (e.g., overseas training/visa scams, art deposit scams, NFT gas fee scams) with our simulated trajectories.
>    These examples illustrate that, at the level of interaction stages, our simulated cases can to some extent reflect key structural elements observed in real-world fraud patterns.

---

### Meta-Review · Area_Chair_5zjz · 2025-12-13

**Summary:**

All reviewers acknowledge the significance of studying collusive fraud in multi-agent systems, with several noting that this is among the first benchmarks addressing coordinated malicious behavior in LLM-based agents. The study yields several interesting findings, such as that LLMs rarely refuse malicious actions in interactive settings, and more capable LLMs can deceive less capable ones.

The reviewers also raise major concerns: 1) the study lacks real-world validity (1Neb, c5hA, U9v1); 2) the agent and social network design may be over-simplified (1Neb, c5hA); 3) the concept of collusion is defined simply as private communication without deep methodological design and experimental analysis (1afb); 4) the proposed mitigation strategies are limited (1Neb, c5hA, U9v1).

**Reviewer Concerns:**

Concerns addressed by rebuttal:
3)

Outstanding concerns:
1), 2), and 4)

**Reviewer Scores:**

1Neb and 1afb haven't responded to the rebuttal. Given that the rebuttal partially addresses their concerns, I think they would keep (or slightly increase) their scores.
c5hA responded to the rebuttal and kept their score (4).
U9v1 increased their score (8) after the rebuttal.

---

### Decision · Program_Chairs · 2026-01-26

Accept (Poster)